# Imaging and Histopathological Analysis of Microvascular Angiogenesis in Photodynamic Therapy for Oral Cancer

**DOI:** 10.3390/cancers15041110

**Published:** 2023-02-09

**Authors:** Tzu-Sen Yang, Yen-Chang Hsiao, Yu-Fan Chiang, Cheng-Jen Chang

**Affiliations:** 1Graduate Institute of Biomedical Optomechatronics, Taipei Medical University, Taipei 110, Taiwan; 2International PhD Program in Biomedical Engineering, Taipei Medical University, Taipei 110, Taiwan; 3School of Dental Technology, Taipei Medical University, Taipei 110, Taiwan; 4Research Center of Biomedical Device, Taipei Medical University, Taipei 110, Taiwan; 5TMU Research Center of Cancer Translational Medicine, Taipei Medical University, Taipei 110, Taiwan; 6Department of Plastic and Reconstructive Surgery, Chang Gung Memorial Hospital, Chang Gung University, Taoyuan City 33302, Taiwan; 7Medical School, University of Queensland, Saint Lucia, QLD 4072, Australia; 8Department of Plastic Surgery, Taipei Medical University Hospital, Taipei Medical University, Taipei 110, Taiwan; 9School of Medicine, College of Medicine, Taipei Medical University, Taipei 110, Taiwan

**Keywords:** photodynamic therapy, chicken chorioallantoic membrane, stimulated malignant oral lesions

## Abstract

**Simple Summary:**

PDT has been recently proposed as a treatment modality for cancers. A histopathological examination of PDT showed that apparent internal hemorrhage and red blood cell (RBC) extravasation are commonly observed in malignant tumors after PDT. Tumor angiogenesis has been identified as one of the key targets in tumor treatments. In this study, in vivo chicken chorioallantoic membranes (CAMs) and a stimulated malignant oral lesions animal model were used. The velocity and structural images of in vivo microcirculation and the relation between the changes in intrinsic signal in blood components and histopathological responses during PDT were examined for more understanding. Our study demonstrated that PDT can target the microvasculatures within the tumor and cause tumor destruction. The optimization of PDT for squamous cell carcinomas of the oral cavity may depend on the ability to cause the selective destruction of only vascular-mediated events or cellular targets without the production of heat through nonthermal mechanisms.

**Abstract:**

The objective of this study is to use imaging and histopathological analysis to characterize and monitor microvascular responses to photodynamic therapy (PDT). In vivo chicken chorioallantoic membranes (CAMs) and a stimulated malignant oral lesions animal model were used to determine the blood flow and the biological activities of Photofrin^®^ (2.5 mg/kg) exposed to different laser power densities at 630 nm. The vascular changes, the velocity of the blood flow, the speckle flow index (SFI) of fluorescence changes, and ultrastructure damage in the microvasculature before and after PDT were recorded. The subcellular localization of Photofrin^®^ revealed satisfactory uptake throughout the cytoplasm of human red blood cells at 10 s and 20 s before PDT. The mean blood-flow velocities of the veins and arteries were 500 ± 40 and 1500 ± 100 μm/s, respectively. A significant decrease in the velocities of the blood flow in the veins and arteries was detected in the CAM model after PDT. The veins and arteries of CAMs, exposed to the power densities of 80, 100, and 120 mW/cm^2^, had average blood-flow velocities of 100 ± 20, 60 ± 10, and 0 μm/s and 300 ± 50, 150 ± 30, and 0 μm/s, respectively. In the stimulated malignant oral lesions animal model, the treated tumors exhibited hemorrhage and red blood cell extravasation after PDT. The oxyhemoglobin and total hemoglobin levels decreased, which resulted in a decrease in tissue oxygen saturation, while the deoxyhemoglobin levels increased. PDT using Photofrin^®^ has the ability to cause the destruction of the targeted microvasculature under nonthermal mechanisms selectively.

## 1. Introduction

Photodynamic therapy (PDT) has been recently proposed as a treatment modality for cutaneous malignancies, which may offer results that are more appealing than those obtained using conventional treatment modalities. The fundamental concept of PDT is that some molecules can function as photosensitizers. The presence of photosensitizers in biological tissue makes the tissue vulnerable to the wavelengths of light absorbed by the chromophore. The stoppage of the blood flow to tumors shortly after the initiation of PDT has been demonstrated; however, the complete cessation of tumor circulation is required to achieve complete eradication. A histopathological examination of PDT showed that apparent internal hemorrhage and red blood cell (RBC) extravasation are commonly observed in tumors after PDT [1,2].

Our previous study used hematopoietic progenitor antigen CD36 to characterize human dermal microvascular endothelial cells (MECs), and the rationale for extending the use of PDT is that PDT facilitates the destruction of targeted blood vessels within the tissue without producing heat [3]. Therefore, the risks related to conventional photothermal effects of high-energy lasers, such as changes in skin pigmentation, atrophy, induration, and hypertrophic scarring, are not expected to occur. For survival, normal cells as well as tumors require oxygen and nutrients that are supplied by the blood. Recently, tumor angiogenesis has been identified as one of the key targets in tumor treatments. In this study, in vivo chicken chorioallantoic membranes (CAMs) and a stimulated malignant oral lesions animal model were used. The velocity and structural images of in vivo microcirculation and the relation between the changes in intrinsic signal in blood components and histopathological responses during PDT were examined.

## 2. Materials and Methods

### 2.1. Photosensitizers

Photofrin^®^ containing porfimer sodium (dihematoporphyrin ether/esters, DHE) is a partially purified preparation of the photoactive ingredients within hematoporphyrin derivative (HPD). Photofrin^®^ obtained from Quadra Logic Technologies, Inc. (Vancouver, BC, Canada) was stored in dark at 4 °C. Photofrin^®^ powder was prepared with 5% dextrose to a final concentration of 2.5 mg/mL. An intraperitoneal (IP) injection of Photofrin^®^ at a concentration of 2.5 mg/kg was administered to the embryo of CAM models [4]. No side effects were observed. Subsequently, 3 h after injecting the embryo with Photofrin^®^, the CAM models were illuminated with a laser beam (λ = 630 nm). An intravenous injection of Photofrin^®^ at a concentration of 2.5 mg/kg was set for the hamsters with stimulated malignant oral lesions. No photosensitivity or side effects were observed. Then, 3 h after intravenously injecting the hamsters with Photofrin^®^, they were treated with a laser beam (λ = 630 nm).

### 2.2. Laser Delivery Systems

Laser illumination was performed with a Coherent Innova 20 argon ion laser (Palo Alto, CA, USA), pumping a Coherent 599-01 dye laser, and was tuned to emit a lower level laser at 630 nm for each experiment. The wavelength was verified using a Jobin Yvon 5/354 UV monochromator (Lonjuneau, France). Illumination was coupled with a 400 mm fused silica fiber optic with a Spectra-Physics Model 316 fiber optic coupler (Mountain View, CA, USA). The output end of the fiber had microlens at its terminal that focused the laser passing into a circular field of uniform light intensity. The laser emitted from the fiber was monitored using a Coherent Model 210 power meter before and after illumination.

### 2.3. In Vivo Chicken Chorioallantoic Membranes (CAM) Model

The chicken chorioallantoic membrane (CAM) is a fitly biological model in this research. Because the CAM microvasculature is located in a transparent matrix, direct viewing and noninvasive imaging of the blood vessels are possible [5,6,7]. The protocol for CAM preparation was a modification of a previously described technique. Fertilized eggs (HylineW36 white leghorn) were washed with 70% alcohol, incubated at 37 °C in 60% humidity, and rolled over hourly. On days 3 and 4 of embryonic development, a hole was drilled in the apex, and 2–3 mL of albumin was aspirated from each egg to create a false air sac. On day 5, after the apex of the chicken eggshell was removed, part of the CAM was exposed by opening a round window (20 mm in diameter) in the shell that was covered with a Petri dish. The eggs were placed in a stationary incubator until the CAM developed completely and was ready for use in our experiment. On days 10–12, sterile Teflon O-rings (inner diameter, 6.2 mm; outer diameter, 9 mm; height, 1.4 mm) were placed on the surface of the CAM for observation. Each demarcated a location where individual blood vessels and capillaries were clearly visible. A drop of normal saline was added within the ring area to reduce light reflection and prevent the desiccation of the CAM during the experiment [8].

Twenty CAMs were given an IP embryo injection of Photofrin^®^ at a concentration of 2.5 g/kg. Subsequently, 3 h after the IP injection of Photofrin^®^, the CAMs were illuminated with a laser beam (λ = 630 nm). Prior to PDT, the window of the CAM was covered with a metal shield with a circular hole (1.5 cm in diameter) that was to be exposed to illumination. The total laser energy density was 100 J/cm^2^ at power densities of 80, 100, and 120 mW/cm^2^. The control groups underwent Photofrin^®^ or laser illumination treatment only. At the time of illumination, the CAM was illuminated with a cold white-light fiber-optic source (model 100 HL, Volpi, Intralux, Amsterdam, The Netherlands) and placed under a stereomicroscope (model SZH, Olympus, Tokyo, Japan) equipped with a video camera (model AC-2510, Panasonic, Osaka, Japan), which provided a magnification of 70× on a color monitor (modelKV-1393R, Sony, Tokyo, Japan).

Recently, laser speckle imaging (LSI), in which coherent light incident on a surface produces a reflected speckle pattern, is related to the underlying movement of optical scatterers, such as red blood cells, indicating blood flow. The corresponding image-processing algorithms can be applied to produce the speckle flow index (SFI) maps of the relative blood flow [9]. In this study, the SFI was used to investigate the blood flow in small animal imaging to characterize the blood flow dynamics associated with the microvasculature. Hence, the intensity of the speckle flow index (SFI) expressed in arbitrary units (a.u.), fluorescence changes, the velocity of the blood flow, and ultrastructure damage in the microvasculature before and after PDT were assessed.

### 2.4. In Vivo Stimulated Malignant Oral Lesions in Animal Model

Twenty-four male Golden Syrian hamsters were included in our experiment. Their age was about 3 months, and body weights ranged between 90 g and 115 g, with a mean weight of 105 g. 0.5% D.M.B.A (9, 10 dimethyl1-1, 2-benzanethracene, Sigma- Aldrich, St Louis, MO, USA) was brushed onto the cheek pouches bilaterally. The stimulated wounds were accomplished using a camel-hair brush. In each animal, both pouches were painted daily for 2 weeks [10]. The images and the biological activities of the stimulated malignant oral lesions were identified via optical coherent tomography (OCT) (OPXION Tech. Inc. Taipei, Taiwan) as well as histopathological confirmation. Here, a super-luminescent diode with a center wavelength of 840 nm was used as the light source for OCT imaging. The stimulated malignant oral lesions were exposed to PDT in a manner similar to our previous study [3]. After two weeks, tumor growths were observed in the pouches of the 24 hamsters. Based on OCT imaging, 24 tumor-bearing hamsters were identified as having squamous cell carcinoma. Of the 20 hamsters, 4 received laser illumination without photosensitizer (control group 1). The other 16 hamsters were given intravenous injections of Photofrin^®^ (2.5 mg/kg body weight). Of these 16 hamsters, 4 received photosensitizers without laser illumination (control group 2). The remaining 12 hamsters received Photofrin^®^ with laser illumination. Then, 3 h after the injection of the photosensitizer, the hamsters were exposed to the laser beam (λ = 630 nm). General lurance-oxygen gas anesthesia was used for all invasive procedures. To increase the specificity of laser illumination on the tumor, a metal shield with a circular hole was used to expose an illuminated area corresponding to the size of each tumor. The total laser energy density was 100 J/cm^2^, with a power of 100 mW at power densities of 100 mW/cm^2^ for each tumor. The 12 hamsters were euthanized immediately (*n* = 3), 3 h (*n* = 3), 6 h (*n* = 3), and 12 h (*n* = 3) after PDT, and the tumors were processed for the evaluation of histopathological destruction by using hematoxylin–eosin staining. Here, five specimens of each tumor were processed.

The correlated intrinsic components such as the tumor oxyhemoglobin (HbO_2_), deoxyhemoglobin (Hb), and total hemoglobin (HbT) concentrations, as well as tissue oxygen saturation (StO_2_), were analyzed in 4 out of the 24 hamsters during tumor growth and after receiving the photosensitizer with laser illumination for PDT. In this study, we applied a typical type of pO_2_ measurement device, the OxyLite system (Oxford Optronics, Oxford, UK), to measure the partial oxygen pressure (pO_2_) in tumor sites, where this method has been applied successfully to many human tumors in the clinic. The animals were treated in accordance with the ARC guidelines stipulated by the Institutional Animal Care and Use Committee at the Chang Gung Memorial Hospital.

## 3. Results

### 3.1. In Vivo Chicken Chorioallantoic Membrane (CAM) Model

Under the fluorescent microscope, and based on the intensity of the speckle flow index (SFI) expressed in arbitrary units (a.u.), the subcellular localization of Photofrin^®^ revealed satisfactory uptake throughout the cytoplasm of RBCs at 10 s and 20 s (Figure 1A,B). The mean blood-flow velocities in the veins and arteries were 500 ± 40 μm/s and 1500 ± 100 μm/s, respectively. The CAMs that received 2.5 mg/kg of Photofrin^®^ and were exposed to laser light at 630 nm at power densities of 80, 100, and 120 mW/cm^2^ had average blood-flow velocities of 100 ± 20, 60 ± 10, and 0 μm/s in the veins, and 300 ± 50, 150 ± 30, and 0 μm/s in the arteries, respectively. The effects of the delivery method on the relative importance of the various treatment variables in PDT were examined in terms of the vascular responses of the CAM system [11]. Because after PDT treatment, no blood flow and no pulsation were observed, we concluded that the imaged vessel was occluded after PDT (Figure 2); that is, the area where the blood vessel color turns white represents the vascular damage induced by PDT. In contrast, the area where the blood vessel color remains unchanged as a control represents the vascular tissue not damaged by PDT.

### 3.2. In Vivo Stimulated Malignant Oral Lesions in Animal Model

Based on OCT imaging, senior pathologists confirmed the histopathological findings of stimulated malignant oral lesions as squamous cell carcinoma through tissue evidence and the received PDT in the 24 hamsters. Figure 3A–C show a typical example of the stimulated squamous cell carcinoma on the buccal mucosa in an OCT image, white-light illumination image, and 3 h after the intravenous application of Photofrin^®^ during the PDT of the tumor, respectively, where the tumor area is marked with the symbol (*). On the other hand, the histopathological findings and fluorescent images (100×) of the stimulated squamous cell carcinoma on the buccal mucosa of the hamsters were identified (Figure 4A,B). In the control animals that received laser only (*n* = 4) or the photosensitizer (*n* = 4) without laser treatment, the treated tumors exhibited minimal hemorrhage and red blood cell extravasation 3 h after PDT. Internal hemorrhage and red blood cell extravasation were observed 6 h after PDT. Severe internal hemorrhage and red blood cell extravasation with the destruction of stimulated squamous cell carcinoma (200×) were observed 12 h after PDT (Figure 4C,D). As the tumor grew, the levels of oxyhemoglobin (HbO_2_) and total hemoglobin (HbT) increased, whereas that of deoxyhemoglobin (Hb) decreased, which resulted in an increase in tissue oxygen saturation (StO_2_). The blood flow in the tumor increased with tumor size growth (Table 1). These results implied that tumor angiogenesis induced an increase in the blood flow and blood volume (HbT), mainly by increasing the flow of arterial blood (HbO_2_). After PDT, the treated tumors exhibited hemorrhage and red blood cell extravasation after PDT. The oxyhemoglobin and total hemoglobin levels decreased, which resulted in a decrease in tissue oxygen saturation, while the deoxyhemoglobin levels increased. Changes in the concentration of the intrinsic components based on time after PDT are shown in Table 2.

## 4. Discussion

Photosensitizers absorb photons of an appropriate wavelength and are promoted to an excited state. The excited photosensitizers subsequently react with a substrate, such as oxygen, to produce highly reactive singlet molecular oxygen that causes irreversible oxidative damage to biologically crucial molecules [12,13,14]. The intersystem transfer from an excited singlet photosensitizer to a triplet state is essential for the production of singlet oxygen (^1^O_2_). Illumination at the appropriate wavelength absorbed by the photosensitizer provides the energy to drive photodynamic reactions without the generation of heat. The provided incident power density is <100 mW/cm^2^. The phototoxic reaction is a local phenomenon that occurs within the same cell on a time scale of microseconds [15]. Optimally, the maximum temperature should be <40 °C to prevent the degeneration of any biological tissue. Efforts to define the mechanisms of PDT action have led to a controversy that attributes cytotoxicity to vascular-mediated events (indirect cell kill) or cellular targets (direct cell kill) of photochemically produced singlet oxygen or other oxygen radicals [11,16].

Photofrin^®^ application caused effective vascular photosensitization, which would possibly result from the cellular events of endothelial cells [17,18,19]. Hence, our experiments were performed to ascertain the time course and dose relationship of Photofrin^®^ in CAMs [8]. Photofrin^®^, as a photosensitizer, was found primarily in the plasma membrane, which suggested that an apparently nonapoptotic mechanism of cell death was involved photochemically [20]. However, Photofrin^®^ located throughout the cytoplasm was highly susceptible to functional inhibition via PDT. We, therefore, believe that PDT-induced damage to mitochondrial function and lysosomes could be the major factor responsible for the effectiveness of PDT [21,22,23,24,25].

In the evaluation of the response of microvasculatures, one of the findings observed in our study was the high vulnerability of arterioles to PDT injury. A possible explanation might be based on considerations of vascular anatomy. This occurred for the three vessel calibers. The arteriolar walls consist of three concentric layers: an endothelial tube, an intermediate layer of smooth muscle cells, and an outer coat of fibrous elements. The thickness of the arteriolar wall varies with vessel caliber and function. On the other hand, the walls of the venules are always thinner than those of arterioles of equal caliber. In PDT, we might anticipate less damage to the thicker, more resilient arteriolar walls. This seems to be consistent with reports of PDT-induced vasoconstriction followed by vasodilatation. However, notably, coagulated blood emboli consisting of agglutinated damaged RBCs can be transported downstream to reduce the blood flow. The occlusion can be in arterioles because blockage occurs before they reach the capillaries and result in a decrease in blood flow and tissue oxygen saturation (StO2) [16]. Permanent clotting is part of our explanation for PDT-induced vascular damage. This difference is apparent under a fluorescent microscope in the subcellular localization of Photofrin^®^ in human RBCs for the analysis of the mean blood-flow velocities in the veins and arteries.

Our previous study identified four PDT-induced vessel damage grades: 0, no observable damage; 1, slight damage, vasodilatation/constriction, and temporary occlusion; 2, moderate damage and permanent occlusion; 3, severe damage, capillary extravasation, and hemorrhage [26]. Because no blood flow and no pulsation were observed after PDT, we concluded that the imaged vessel was occluded. Meanwhile, the effects of the delivery method on the relative importance of the severity of the various treatment variables in PDT have been previously examined in terms of the vascular coagulation of the CAM system and stimulated malignant oral lesions [26,27,28].

Hyperkeratosis cases have risen with a prevalence of 1–4% in the general population in Asia. Malignant transformation occurs in approximately 6% of lesions over 5 years, depending on the lesion type. The standard treatment for early-stage (AICC Stage I and II) primary mucosal malignancies of the upper aerodigestive tract (UADT) is surgery and/or radiotherapy. Recently, promising results have been reported using PDT on complicated malignant lesions. The in vivo stimulated malignant oral lesions model in our study demonstrated that the blood flow in the tumors increased as the tumors grew. As the tumors grew, oxyhemoglobin (HbO_2_) and total hemoglobin (HbT) increased, while deoxyhemoglobin (Hb) decreased, which resulted in an increase in tissue oxygen saturation (StO_2_). These results implied that the tumor angiogenesis induced an increase in the blood flow and blood volume (HbT), mainly by increasing the arterial blood (HbO_2_). After PDT, marked PDT-induced destruction vasculature with hemorrhage and RBC extravasation was observed in our histopathological examinations. These findings were consistent with the changes in the concentration of intrinsic components based on time after PDT (Table 2). Our results demonstrated that Photofrin^®^ has the ability to cause the destruction of microvasculature in CAM models and to stimulate malignant oral lesions. Photosensitizers with greater tissue penetration of longer wavelengths make Photofrin^®^ ideal for the treatment of large, deep malignant tumors in PDT. Advancements in this field will enable the use of PDT for treating patients with oral cancer and other malignant neoplasms and, thus, may expand the population of patients expected to benefit from PDT [10]. In addition, the rationale for extending the use of PDT to hemangiomas, hypervascularity cutaneous anomalies, and deep, large, or complicated vascular tumors is possible.

## 5. Conclusions

In conclusion, PDT using Photofrin^®^ can target the microvasculature within the tumor and cause tumor destruction. Continuing improvement in treatment results will depend on the ability to cause the selective destruction of only vascular-mediated events or cellular targets without the production of heat through nonthermal mechanisms. The optimization of PDT for squamous cell carcinomas of the oral cavity may extend to more fields for the application of PDT to tumors in other anatomic areas or of other histopathological patterns.

## Figures and Tables

**Figure 1 cancers-15-01110-f001:**
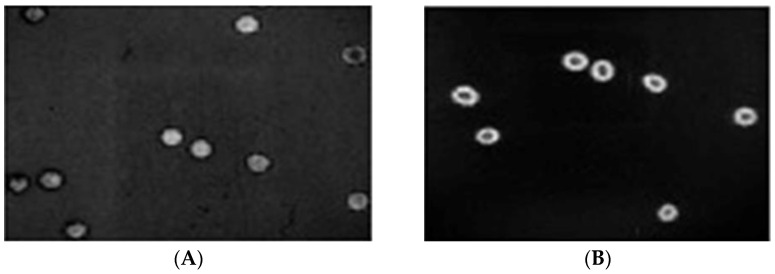
Under the fluorescent microscope (40×), the subcellular localization of Photofrin^®^ revealed satisfactory uptake throughout the cytoplasm of CAM RBCs at 10 s and 20 s (**A**,**B**).

**Figure 2 cancers-15-01110-f002:**
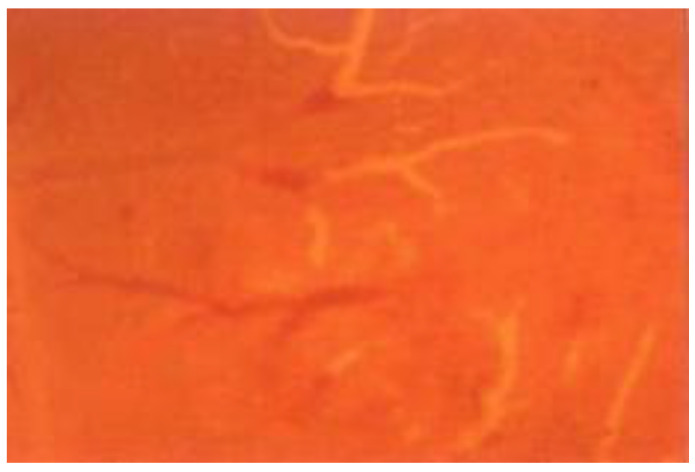
The PDT-induced vascular damage was evaluated, and observable damage with vascular constriction or occlusion was evaluated under a stereomicroscope. The area where the blood vessel color turns white represents the vascular damage induced by PDT (right-hand side). In contrast, the area where the blood vessel color remains unchanged as a control represents the vascular tissue not damaged by PDT (left-hand side).

**Figure 3 cancers-15-01110-f003:**
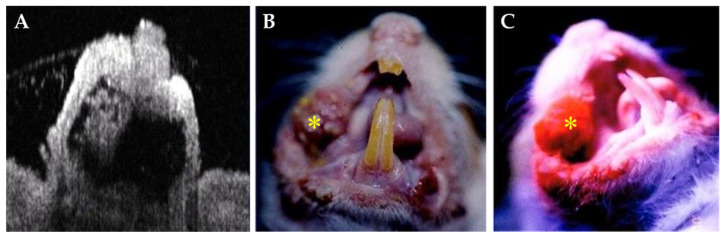
An example of stimulated squamous cell carcinoma on the buccal mucosa in image of OCT concept (**A**), under white light (**B**), and 3 h after intravenous application of Photofrin^®^ during PDT of tumor (**C**); the tumor area is marked with the symbol (*).

**Figure 4 cancers-15-01110-f004:**
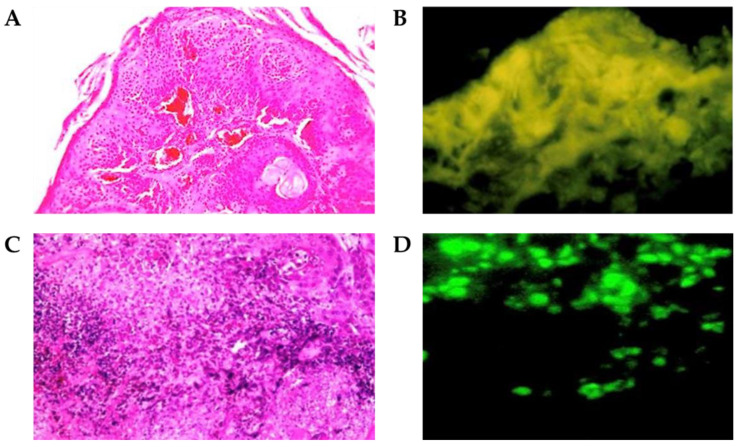
The histopathological findings and fluorescent images (100×) of stimulated squamous cell carcinoma on the buccal mucosa of hamsters (**A**,**B**) were identified. Severe internal hemorrhage and red blood cell extravasation with the destruction of the tumor 12 h after PDT were demonstrated (**C**,**D**) (200×).

**Table 1 cancers-15-01110-t001:** Mean changes in intrinsic signals based on time during tumor growth.

Day (s)	0	2nd	4th	6th	8th	10th	12th
HbO_2_(μM)	63	80	109	123	159	210	215
Hb (μM)	57	52	47	40	38	36	35
HbT (μM)	120	132	138	163	197	246	250
StO_2_(%)	38	45	54	62	72	89	92

HbO_2_—oxyhemoglobin; Hb—deoxyhemoglobin; HbT—total hemoglobin; StO_2_—tissue oxygen saturation.

**Table 2 cancers-15-01110-t002:** Mean changes in intrinsic signals based on time after photodynamic therapy.

Day (s)	* 14th	16th	18th	20th	22nd	24th	26th
HbO_2_ (μM)	216	160	130	117	91	84	78
Hb (μM)	36	45	51	57	65	69	72
HbT (μM)	252	220	200	187	173	168	150
StO_2_ (%)	95	86	72	65	60	55	47

HbO_2_—oxyhemoglobin; Hb—deoxyhemoglobin; HbT—total hemoglobin; StO_2_—tissue oxygen saturation; *—photodynamic therapy (PDT).

## Data Availability

The data presented in this study are available in this article.

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
