# Peer review of "Imaging and Histopathological Analysis of Microvascular Angiogenesis in Photodynamic Therapy for Oral Cancer"

_cancers, 2023, doi:10.3390/cancers15041110_

Round 1

Reviewer 1 Report

Authors in the manuscript showed that the OSCC hamster and CAM model on PDT results in selective microvasculature damage, which is one of the factors of successful PDT. This is the work to look at the insight of in-vivo post-PDT effects on oral lesions. Histopathological and blood biomarkers were also analyzed here.  

There are some comments that need to address. 

Major comments:

1.     Chorioallantoic membrane (CAM) model refs for angiogenesis experimentation. Add the most recent references using CAM in the field of PDT.

2.     By what method was tumor oxyhemoglobin quantified?

3.     In Figure 1, where is the image with control and treated PDT and marking of damaged vascular damage?

4.     Figures 3, and 4. rewrite panels A, B, and C and mark the tumor area, is it pre and post-PDT images?

5.     This may be interesting to have some correlation graph of quantified biomarkers in the OSSC hamster and CAM model. 

Minor comments:

1.     What is speckle flow index (SFI)

Author Response

The authors would like to thank Reviewer #1 for your helpful comments that allowed us to revise our manuscript.

Reviewer 2 Report

The reviewed article is devoted to the analysis of the mechanisms of damage to the vascular system as a result of photodynamic therapy. It should be noted that although the involvement of the vascular system of tissues in photodynamic effects has been discussed for 40-50 years, the exact mechanisms of the vascular system's participation in the tissue response to PDT are not yet clear. The article presents interesting results illustrating changes in the speed of blood flow in blood vessels, the degree of oxygen supply to tissues, and violations of the integrity of the walls of blood vessels. These results were obtained using two experimental models. In general, the article presents a relatively new approach to the study of the problem, the results obtained will be of interest to researchers specializing in the development of photodynamic therapy methods. The idea of the article and its discussion are described clearly. In my opinion, the experimental techniques used in the study are not fully presented. The description of the results obtained should also be expanded, an explanation should be given about the relationship of those parameters that are determined during the study with changes in the structural and functional characteristics of tissues. After reading the article, a question arises. As far as it is possible to compare the results for the two models. With the IP introduction of a photosensitizer, the kinetics of changes in its concentration in the vascular system is mediated by the process of transfer into the blood. This can lead to significant shifts in both the level and time dependence of the photosensitizer concentration in the blood. The photosensitizer used is a mixture of various photoactive components. Different rates of penetration of components into the blood can also significantly change the dynamics of photosensitizer bio-distribution compared to intravenous administration.

A few more questions and comments on the design of the article.

1. how was the photosensitizer binding evaluated in blood cells and in the endothelium?

2. what is the cause of dramatic changes in blood flow parameters at the highest radiation flow density?

3. does the use of optical measurement methods, coupled with additional irradiation of samples, lead to a change in the intensity of tissue response to photo-action?

4. a different form of representation of numerical results should be given. In the article they are presented in the form X ± y, what does this mean?

Author Response

The authors would like to thank Reviewer #2 for your helpful comments that allowed us to revise our manuscript.

Reviewer 3 Report

Comments: Would you please clarify the following arguments before acceptance?

1.     In introduction section , it was written "the destruction of targeted blood vessels". However, there was no explanation for how  these materials can be targeted to blood vessels.

2.     In Figure 3, manage the labels of figures

3.     In Figure 4, A (character) was  duplicated

4.     Visualization of blood vessels using immunohistochemical staining should be obtained

5.     It was written in discussion  section" PDT induced damage to mitochondrial function and liposomes" . The authors didn't explain what does mean liposome here?.

Author Response

The authors would like to thank Reviewer #3 for your helpful comments that allowed us to revise our manuscript.

Reviewer 4 Report

The manuscript "Imaging and Histopathological Analysis of Microvascular Angiogenesis in Photodynamic Therapy for Oral Cancer" describes the results of 2 in vivo models investigating the effect of Photofrin®- induced vascular changes, hemorrhage, RBC extravasation, and resultant oxyhemoglobin, total hemoglobin, tissue oxygen saturation, and deoxyhemoglobin level changes.

In general, I find that the topic of this research is interesting, and that the in vivo models are good to investigate the questions the authors had. The manuscript itself, however, is not well-written, with lacking information on some methods, and in general, it just doesn`t flow well - with a lot of sentences not connected. I would suggest a major rewriting of the manuscript with possibly some additional information needed to be added (see below).

In detail:

1. For the CAM model: the authors state that human rbc are shown (Figure 1.) and are used. It is confusing to me: where and why did you use human rbcs in this study for basic specle flow measurement, and not animal rbcs (chicken, and/or hamster) from animals used in this study? Did you check the uptake to these rbcs? If not, please check it and include the data in the revised manuscript.

2. Methods questions:

- How were tumor oxyhemoglobin (HbO2), deoxyhemoglobin (Hb), and total hemoglobin (HbT), and tissue oxygen saturation (StO2) measured and analyzed? They are not detailed.

- Section 2.4.: How did you analyze the H&E images? how many slides were taken, and from what areas from one animal? How many areas did you analyze for one section and for one animal? What did you actually look at and what did you score and how on the images? How did you measure the "destruction"? Please describe this in more details.

- Section 3.1.: confusing description, and most here should be part of the methods. It is also not well-described here what you call slight damage etc. Please clarify. Also, the actual results should be highlighted, it is pretty confusing as it is now.

3. In general, the manuscript is not well-written (ie. "General is of lurance-oxygen gas anesthesia was used for all invasive procedures" in chapter 2,4,), and it doesn`t flow, especially in the simple summary, abstract, introduction, 2.1., 2.4., 3.1., and discussion chapters. Please revise it.

4. minor correction for Figure 1 is needed: please adjust the letters on the pictures to one of the corners, and delete the ¶ signs if not applicable.

Author Response

The authors would like to thank Reviewer #4 for your helpful comments that allowed us to revise our manuscript.

Round 2

Reviewer 3 Report

The manuscript was revised point by point according to reviewer's comments and It can be accepted.

Author Response

The authors would like to thank Reviewer #3 for his or her careful review of our manuscript and for providing us with constructive comments and suggestions to further improve the quality of our manuscript. 

Reviewer 4 Report

The manuscript significantly improved since the first submission. In thos form, with further minor changes, I suggest it to be accepted.

The methods section is largely improved as well, I would still be interested to see how many areas wnd with what magnification the H&E pictures were taken. I think this information should be added before publishing.

Minor re-reading is advised, ie to correct Introduction,  which starts with saying "PDT", explaining the meaning later. But in general, the manuscript is easy to read and understand now.

Author Response

The authors would like to thank Reviewer #4 for his or her careful review of our manuscript and for providing us with constructive comments and suggestions to further improve the quality of our manuscript. 
